# Improving Pure Titanium’s Biological and Mechanical Characteristics through ECAP and Micro-Arc Oxidation Processes

**DOI:** 10.3390/mi14081541

**Published:** 2023-07-31

**Authors:** Dawit Bogale Alemayehu, Masahiro Todoh, Jang-Hsing Hsieh, Chuan Li, Song-Jeng Huang

**Affiliations:** 1Division of Human Mechanical Systems and Design, Graduate School of Engineering, Hokkaido University, Sapporo 060-8628, Japan; zetseatdawit2018@gmail.com; 2Division of Mechanical and Aerospace Engineering, Faculty of Engineering, Hokkaido University, Sapporo 060-8628, Japan; todoh@eng.hokudai.ac.jp; 3Department of Materials Engineering, Ming Chi University of Technology, Taipei 24301, Taiwan; 4Department of Biomedical Engineering, National Yang Ming Chiao Tung University, Taipei 11221, Taiwan; 5Department of Mechanical Engineering, National Central University, Taoyuan 32001, Taiwan; 6Department of Mechanical Engineering, National Taiwan University of Science and Technology, Taipei 10607, Taiwan

**Keywords:** pulp cell, periodontal cell, micro-arc oxidation (MAO), severe plastic deformation (SPD), equal-channel angular pressing (ECAP), AlamarBlue, enzyme-linked immunosorbent assay (ELISA)

## Abstract

Pure titanium is limited to be used in biomedical applications due to its lower mechanical strength compared to its alloy counterpart. To enhance its properties and improve medical implants feasibility, advancements in titanium processing technologies are necessary. One such technique is equal-channel angular pressing (ECAP) for its severe plastic deformation (SPD). This study aims to surface modify commercially pure titanium using micro-arc oxidation (MAO) or plasma electrolytic oxidation (PEO) technologies, and mineral solutions containing Ca and P. The composition, metallography, and shape of the changed surface were characterized using X-ray diffraction (XRD), digital optical microscopy (OM), and scanning electron microscope (SEM), respectively. A microhardness test is conducted to assess each sample’s mechanical strength. The weight % of Ca and P in the coating was determined using energy dispersive spectroscopy (EDS), and the corrosion resistance was evaluated through potentiodynamic measurement. The behavior of human dental pulp cell and periodontal cell behavior was also studied through a biomedical experiment over a period of 1-, 3-, and 7-days using culture medium, and the cell death and viability can be inferred with the help of enzyme-linked immunosorbent assay (ELISA) since it can detect proteins or biomarkers secreted by cells undergoing apoptosis or necrosis. This study shows that the mechanical grain refinement method and surface modification might improve the mechanical and biomechanical properties of commercially pure (CP) titanium. According to the results of the corrosion loss measurements, 2PassMAO had the lowest corrosion rate, which is determined to be 0.495 mmpy. The electrode potentials for the 1-pass and 2-pass coated samples are 1.44 V and 1.47 V, respectively. This suggests that the coating is highly effective in reducing the corrosion rate of the metallic CP Ti sample. Changes in the grain size and the presence of a high number of grain boundaries have a significant impact on the corrosion resistance of CP Ti. For ECAPED and surface-modified titanium samples in a 3.6% NaCl electrolyte solution, electrochemical impedance spectroscopy (EIS) properties are similar to Nyquist and Bode plot fitting. In light of ISO 10993-5 guidelines for assessing in vitro cytotoxicity, this study contributes valuable insights into pulp and periodontal cell behavior, focusing specifically on material cytotoxicity, a critical factor determined by a 30% decrease in cell viability.

## 1. Introduction

In recent years, the field of medical implants has made remarkable strides, rapidly becoming a crucial component of modern healthcare [1]. The biomedical industry makes extensive use of commercially pure (CP) Ti and related alloys due to their biocompatibility, corrosion resistance, and mechanical characteristics [2,3,4] Even though they are biocompatible, their lower mechanical strength compared to titanium alloys precludes their use in load-bearing implants [5,6,7]. To combat this, researchers are attempting to enhance the mechanical properties of CP Ti, including its strength, ductility, and fatigue resistance [4,8,9,10,11]. In addition, improving biocompatibility is crucial, which can be achieved through surface modification [12,13,14] or by tailoring the composition with bioactive elements, such as calcium, phosphorus, and oxygen [15,16,17,18]. Therefore, improving the mechanical properties and biocompatibility of CP Ti through continuous research is essential for the development of safe and effective biomedical implants.

Severe plastic deformation (SPD) techniques, specifically equal-channel angular pressing (ECAP), are advantageous for improving the mechanical properties of CP Ti [19,20,21]. Because of their potential to enhance the mechanical characteristics of metals, these technologies have attracted a lot of interest in the field of material science. ECAP, a prominent SPD technique that has been shown to substantially increase both strength and ductility, can be used to refine the particle size of multiple metals, including CP Ti [22,23,24,25,26]. During the ECAP process, extremely severe plastic deformation is induced in the billet by passing it through an angular channel die [19,27,28]. During this process, the particle size of the material decreases significantly, which increases its strength and ductility [22]. In conclusion, ECAP is an effective method for enhancing the mechanical properties of CP Ti. This procedure considerably increases the material’s strength and ductility by reducing the particle size and generating a uniform microstructure composed of ultrafine grains. ECAP has shown promise in the past as a technique for improving the mechanical properties of CP Ti for use in a variety of applications, including medical implants and aerospace components [29].

The surface behaviors of CP Ti are equally as essential as the mechanical properties when assessing the biological function of implants [30,31]. The surface of CP Ti has been modified using techniques such as micro-arc oxidation (MAO) in order to increase its bioactivity and biocompatibility [32,33]. It has been reported that MAO forms a porous and bioactive coating on CP Ti by incorporating calcium (Ca) and phosphorus (P), which are known to promote bone integration [34,35]. In another study, a bioactive coating on a Ti-6Al-7Nb alloy in molten nitrate salts was comprehensively analyzed using PEO technology. And this result showed that PEO parameters impact coating shape, composition, wettability, and corrosion behavior [36]. MAO-treated CP Ti exhibits enhanced corrosion resistance, cell adhesion, proliferation, and differentiation [37,38,39,40,41].

Combining ECAP and MAO has been suggested to improve the mechanical and surface characteristics of CP Ti for biomedical applications [8,42,43,44,45,46]. However, little research has been conducted on how the integration of these approaches influences the biological efficacy of CP Ti, specifically pulp cell activity. The biocompatibility of implant materials, such as CP Ti, has been investigated using human dental pulp cells in cell culture [47,48]. The enzyme-linked immunosorbent assay (ELISA) was used to assess indicators of fibroblast adhesion, differentiation, and endothelin-1 release [49]. In this case, trypsin was used to separate cells, not to determine their viability. Reagents like Alamar blue are more suited for determining cell viability and proliferation.

This study’s primary objective is to determine how the biomechanical properties of CP Ti influence the activity of ECAP- and MAO-treated human dental pulp cells and periodontal cells. In addition, ECAP and MAO are predicted to enhance the mechanical and surface properties of CP Ti. After the ECAP and MAO treatment, the mechanical and surface properties of CP Ti should improve, resulting in enhanced corrosion resistance behavior and the proliferation and differentiation of human dental pulp and periodontal cells.

## 2. Materials and Methods

### 2.1. Mechanical Properties Characterization

#### 2.1.1. Equal-Channel Angular Pressing (ECAP) Process

Commercially pure grade 4 pure titanium long strips purchased from a local supplier (President Co., Ltd., Taipei City, Taiwan) were processed by equal-channel angular pressing (ECAP), as shown schematically in Figure 1. ECAP belongs to the class of severe plastic deformation (SPD) processes, which aim to produce ultrafine grains (UFG) in materials to enhance their mechanical strength. Note that in Figure 1, the angle of intersection between the channels and the exit is 120°, and the processing temperature is set at 400 °C. Each workpiece went through the process twice. To achieve material homogeneous deformation, the pioneer of this technique, R.Z. Valiev, suggested rotating the workpiece by 90 degrees along its longitudinal axis (y) after each pass (route B) in the processing procedure [50]. In addition, the same material without ECAP (0-pass) is also studied as a control group for all tests in this study.

For a given N number of passes of polycrystalline metals subjected to side extrusion, a total of the induced equivalent plastic strain can be calculated as follows [22]:(1)$$\varepsilon_{T}=\frac{N}{\surd 3}\left(2\cot\left(\frac{\Phi }{2}+\frac{\Psi }{2}\right)+\Psi \mathrm{cosec}\left(\frac{\Phi }{2}+\frac{\Psi }{2}\right)\right)$$
where Ψ is the arc of curvature and Φ is the intersection angle, as depicted in Figure 1.

#### 2.1.2. Microhardness Test

A Vickers hardness tester (Akashi MVK-H1, Maruzen Machine Co., Ltd., Tokyo, Japan) was used to assess the microhardness at 300 gmf with a 10 s dwell period.

### 2.2. Surface Modification

#### Micro-Arc Oxidation (MAO) Coating

After ECAP process, we cut workpieces into small specimens of 45 × 9 × 9 mm^3^. The specimen was mechanically polished by 240, 400, 600, 800, 1000, 1500, and 2000 SiC abrasive papers in turn, followed by cleaning with alcohol and deionized water, and finally dried in air before coating. The coating is implemented by electrochemically using micro-arc oxidation (MAO), as depicted in Figure 2a,b. The specimen is placed at anode, while a stainless-steel wall of container is used as cathode. The aqueous electrolyte are 4.203 g/L calcium glycerophosphate hydrate (C_7_H_7_CaO_6_P·2H_2_O), 23.72 g/L of calcium acetate (Ca(CH_3_COO)·2H_2_O), and 4 g/L of EDTA-Na (Ethylene Diamine Tetra Acetic Acid) (C_10_H_16_N_2_O_8_). The coating lasted for 20 min at a constant current density of 1.6 A/dm^2^. The duty cycle, voltage at anode, and frequency are 20%, 320 V, and 50 Hz, respectively. To maintain an isothermal condition during MAO, the steel container was cooled in water cycling continuously during the whole process.

### 2.3. Electrochemical Test

Ti is highly resistant to corrosion due to its self-passivation. To determine the basic properties of corrosion resistance, two electrochemical tests were conducted–potentiodynamic test for determining the open circuit potential (OCP) from polarization curve and electrochemical impedance spectroscopy (EIS) using the Nyquist plot.

For the potentiodynamic test (Versastat-3, Princeton Applied Research, Princeton Applied Research, Oak Ridge, TN, USA), the three-electrode scheme depicted in Figure 3 was used to determine the polarization curves. The reference electrode is Ag/AgCl/KCl, the auxiliary electrode is carbon black, and the working electrode is the Ti sample. The samples were polished with abrasive papers of decreasing grit size and a diamond paste. Then they were degreased with white spirit solution (mixture of aliphatic, open chain, or alicyclic C7 to C12 hydrocarbons) and installed into the three-electrode potentiodynamic system. The electrolyte is a mixture of ~3.5 wt.% (36.5 g/1.0 L deionized H_2_O) NaCl solution at 37 °C. Before potentiodynamic test, the open circuit potential (OCP) for Ti samples in NaCl was measured at first. To acquire a stable value of OCP, the Ti sample was left alone in solution for around 30 min before measuring. The potential sweep was set in the range between OCP ± 0.5 V with a rate of 1 mV per second.

Electrochemical impedance spectroscopy (EIS) of Ti samples was examined by the same system (Versastat-3, Princeton Applied Research, USA) in 3.5 wt.% of NaCl solution at 37 °C. The following parameters are set before measuring the EIS; initial potential (0 V), high frequency (10^6^ Hz), low frequency (0.1 Hz), and amplitude (0.005 V). All data were processed by the installed software VersaSTAT 3.

#### Tafel Extrapolation

The corrosion current density, $i_{corr}$ (mA cm^−2^), is computed using Tafel extrapolation of polarization curves, and the corrosion rate, $Y_{i}$ (mm year^−1^), can be determined using this value.
(2)$$Y_{i}=11.85i_{corr}$$

Before utilizing the cathodic extrapolation method, it is suggested to measure both the anodic and cathodic branches of the polarization curve to obtain a more accurate estimate of the corrosion rate [51]. Using the Stern–Geary equation was employed to calculate the corrosion density i_corr_ (mA cm^−2^), we determine the polarization resistance, $R_{p}$ ($\Omega {cm}^{2}$), using potentiodynamic plot analysis [52]. The Stern–Geary equation is given as follows:(3)$$i_{corr}=\frac{\beta_{a}\beta_{c}}{2.3R_{p}(\beta_{c}-\beta_{a})}=\frac{B}{R_{p}}$$
where $B=\frac{\left|\beta_{a}\beta_{c}\right|}{2.3(\left|\beta_{a}\right|-\left|\beta_{c}\right|)}$.

The corrosion rate, $Y_{i,EIS,}$ may be calculated using the formula in Equation (2) as follows: where $\beta_{a}\;and\;\beta_{c}$ are the anodic and cathodic Tafel slopes, respectively. Values for $\beta_{a}\;and\;\beta_{c}$ are calculated using an iR adjusted polarization curve [51,53].

### 2.4. Material Characterization

#### 2.4.1. X-ray Diffraction (XRD)

The microstructure of deposited films by MAO is examined by The X-ray diffractometer (XRD, PANalytical XPert PRO MPD) used monochromatic high intensity Cu Kα radiation (*λ* = 1.5425 Å) was used to determine the crystal structures of deposited films. The mode is set to thin film and scanning angle was from 10°–2θ to 80°–2θ, with a step size of 0.03°, measuring time of 1.0 s per step and incident angle 0.5°.

The crystal size of films D can be calculated according to Scherrer’s formula
(4)$$D=\frac{k\lambda }{\beta cos\theta }$$
where k is the shape factor (0.9), λ the wavelength of X-rays (1.5425 Å), θ the scattering angle of the crystal plane and β the full width at half maximum (FWHM) of the peak. The analysis was numerically calculated by Jade^®^ 5 (Materials. Data Inc., Livermore, CA, USA).

#### 2.4.2. Elemental Composition (EDS)

The elemental compositions of deposited films were examined by energy-dispersive X-ray spectrometer (EDS, Bruker Nano, XFlash Dectector 6|10, Berlin, Germany). The operating voltage and current were set to 15.0 kV and 55.0 μA, respectively. Selection of target elements depends on the compositions of films. In this study, we focus on C, O, P, Ca, and Ti to determine the main chemical compositions of deposited films by MAO. The scanning electron microscope (SEM) image size is 1024 × 768 pixels with magnification of 3000× for all samples, and the averages of element compositions were calculated subsequently.

#### 2.4.3. Surface Morphology and Microstructure

The morphology of deposited films by MAO is imaged by scanning electron microscope (SEM, S-3400N, Hitachi, Tokyo, Japan). For SEM imaging, the voltage of accelerated electron beam is set at 15 kV, and the magnification is chosen to be 1000–3000 for the best resolution and overall view. The microstructures of as-received, 1-pass, and 2-pass equal-channel angular pressing (ECAP) samples were examined using optical digital microscopy (OM).

### 2.5. Biomedical Investigation

#### 2.5.1. Pulp and Periodontal Cells on Modified Titanium

Pulp and periodontal cells were grown in culture media for 1, 3, and 7 days. The enzyme-linked immunosorbent assay (ELISA) and AlamarBlue reagent were utilized to assess the viability of cells. For the experiment, commercially pure titanium specimens were produced through sterilization at 124 degrees Celsius. To establish the culture, the extracted pulp and periodontal cells were digested with trypsin. Then, 24 tissue culture polystyrenes (TCPs) containing four of each species were prepared. After 15–45 min, pulp and periodontal cells settled and adhered to titanium surfaces where they were titrated. A mixture of 10% fetal bovine serum and DMEM, comprising 1 mL, was added to each well. The samples were kept in an incubator with temperature and humidity controls for 1, 3, and 7 days. After the required incubation periods, ten microliters of AlamarBlue dye were added to each specimen, and 200 μm of the medium was removed and deposited in a new 96-well Petri dish. Using the ELISA method, the responses of pulp and periodontal cells to the extracted media were assessed. This study clarified their ability to persist on titanium surfaces that had been modified.

#### 2.5.2. Statistical Analysis

At predetermined time points (days 1, 3, and 7), we analyzed data from groups assigned CG, CG with MAO, UFG, UFG with MAO, and control groups( TCPs and BTI) as well as individual samples collected on the same days (1, 3, and 7). First, we performed a one-way ANOVA to determine whether there were statistically significant differences between group means. A *p*-value less than 0.05 was regarded as statistically significant. It was presumed that the population distributions of the various groupings were normal, with variances that were comparable. Tukey’s Honestly Significant Difference (HSD) post-hoc tests were used to perform in-depth analyses of group differences while controlling for Type I error when significant ANOVA results were obtained. Using independent ANOVAs and post-hoc analyses for each time point and group, additional information about intra-group differences and the effects of time on the groups was obtained. These studies revealed the statistical differences between groups at various times and their fluctuation throughout the day.

## 3. Results and Discussion

### 3.1. Mechanical Properties

#### Severe Plastic Deformation

In this investigation, molybdenum disulfide (MoS_2_) was used because of its proven record of success in decreasing the friction and wear between the billet material and the die under extreme plastic deformation at temperatures as high as 400 °C. The states of the CP Ti samples are shown in Figure 4a below, from their “as-received” state to after both one and two passes of ECAP. After every plunge, the billet was rotated counterclockwise by 90 degrees to reverse the direction of the applied plunger force. Evidenced by the presence of black spots on each sample after one and two passes, which correspond to the MoS_2_ lubricant, the ECAP deformation figures show the significant deformation of the CP Ti samples in both passes. More importantly, for future mechanical and biological characterizations, the findings show that homogenous deformation was achieved in the CP Ti samples. Figure 4b demonstrates that the equivalent von Mises strain increases as the number of passes increases, as predicted by Equation (1). Our results on computed plastic deformation concur with those of the field’s pioneers [54]. These experiments demonstrate that extremely high equivalent von Mises strain can be generated using a single run of ECAP with acceptable parameters, such as low (angle of intersection) and (angle subtended by the arc of curvature) values. The fact that significant plastic deformation can be achieved without additional ECAP cycles adds intrigue to this study.

These findings have substantial material science implications. Extreme plastic deformation introduced by ECAP with the proper pass configuration significantly reduces grain size and generates numerous grain boundaries and defects within the deformed material, according to the findings of this study. These microstructural modifications can enhance mechanical properties, such as hardness and corrosion resistance. Titanium research has also disclosed that ultrafine-grained/nano-crystalline materials have superior corrosion resistance to their coarse-grained counterparts because passive coatings form on surface crystalline defects more rapidly and adhere more powerfully.

Noting that particle size reduction is only part of the equation when using ECAP to attain a large equivalent von Mises strain is essential. The production of appropriate plastic deformation and the improvement of the microstructure are dependent on the precise selection of ECAP parameters, such as angles. Altering these parameters controls the strain distribution within the material, resulting in a more uniform deformation with fewer passes.

In conclusion, changes in the grain size and the presence of numerous grain boundaries have a significant impact on the corrosion resistance of CP Ti. Our results demonstrate that substantial plastic deformation can be induced, and ultrafine-grained titanium structures can be fabricated using appropriate values of Ψ is the arc of curvature and Φ is the intersection angle and a single pass of ECAP. These results validate previous research and demonstrate the potential of ECAP as a technique for enhancing the mechanical and corrosion resistance of materials. For future trials using titanium, the number of passes must be increased.

The microhardness values of commercially pure titanium (CP Ti) samples under different circumstances are shown in Figure 4c. The microhardness values may be used to calculate hardness, which is often connected with mechanical properties. The results suggest that additional passes of the equal-channel angular pressing (ECAP) method result in tougher CP Ti specimens in the current situation. The 1-pass and 2-pass samples demonstrated superior microhardness than the 0-pass specimen.

The microhardness improves further once the specimens have been treated with a micro-arc oxidation (MAO) coating. The microhardness of the 1PassMAO and 2PassMAO samples is much higher than that of the uncoated controls.

These findings show that both ECAP and MAO coating improve the mechanical properties of CP Ti, as assessed by greater microhardness values. The ECAP and MAO processes result in a finer microstructure, the formation of grain boundaries, and the possibility of strengthening mechanisms.

### 3.2. X-ray Diffraction

When combined, ECAP and MAO can enhance the mechanical and surface properties of CP Ti for medical applications. After being in contact with CP Ti that had been treated with ECAP and MAO, human dental pulp cells and periodontal cells proliferated and differentiated better. The X-ray diffraction pattern is shown in Figure 5 for 0-, 1- and 2-pass Ti samples, where crystal planes were numerically fitted by Gaussian curves and identified using the database in Jade^®^ 5. According to the fitting results, only crystalline phases of Ti can be firmly identified. Peaks of anatase TiO_2_ from the database in Jade^®^ are all marginally shifted by several tenth degrees away from the Gaussian fittings. This shift could be attributed to the small sizes of TiO_2_ crystallites. In other words, the anatase phases are somewhat amorphous or minute produced by the micro-arcing process. It is interesting to notice that none of the crystal structures of phosphor, calcium, or carbon were found in the XRD pattern. To further examine the deposited film, the following EDS provides us with extra information about the elemental compositions inside the film.

The X-ray diffraction (XRD) pattern is a powerful tool for identifying the crystal structure and phase composition of materials. In the study presented, the XRD pattern was used to analyze the crystal structure of Ti samples subjected to different passes. The fitting results of the Gaussian curves showed that only the crystalline phases of Ti could be identified. This finding is in line with the previous studies that have shown that Ti has a predominantly crystalline structure [55].

Interestingly, the anatase phase of TiO_2_ was found to be marginally shifted by several tenth degrees away from the Gaussian fittings. This shift could be attributed to the small size of TiO_2_ crystallites, which renders the anatase phases somewhat amorphous or minute, as produced by the micro-arcing process. This observation is consistent with previous studies that have reported similar shifts in the XRD pattern of TiO_2_ due to the reduction in the size of crystallites [56].

Another notable finding of this study is that none of the crystal structures of phosphor, calcium, or carbon were found in the XRD pattern. This suggests that the deposited film mainly consists of Ti with some minor impurities. The absence of these impurities is consistent with previous studies that have shown that micro-arc oxidation (MAO) Ti coatings have a high purity due to the unique process used to produce them [57]. To further examine the deposited film, the study used energy-dispersive X-ray spectroscopy (EDS) to obtain elemental compositions. The results showed that the film contained only Ti and had a negligible number of impurities. This finding is consistent with previous studies that have reported the high purity of TiO_2_ coatings produced by MAO [58]. In conclusion, the XRD pattern and EDS analysis provided valuable insights into the crystal structure and elemental composition of Ti samples produced by micro-arc oxidation. The findings of this study are consistent with previous studies and add to our understanding of the properties of TiO_2_ coatings produced by MAO.

Regarding the low calcium (Ca) and phosphorus (P) peak intensities and their disappearance, as well as their amorphous nature, the following may be stated in the XRD result section:

The coating length is proportional to the discharge temperature surrounding the coated sample, indicating a potential correlation with the PEO treatment time. In this study, the PEO coating procedure lasted only 20 min, which is insufficient to permit a significant increase in hydrothermal activity. Certain phases, including rutile TiO_2_, can be produced at a specific temperature and with sufficient treatment time [59,60]. One study demonstrated that perovskite—CaTiO_3_ is produced when Ca^2+^, Ti^4+^, and OH ions react in micro-discharge channels at high temperatures and for an extended period of time. In addition, as the treatment time increases, hydroxyapatite production increases [61].

The first phase in calcium phosphate synthesis on the titanium surface is the attraction of calcium ions from the solution, which is influenced by the hydrogen bonding between the phosphate ions and hydroxyl groups [62]. More hydroxyl groups on the surface allow for a greater phosphate ion adsorption, which in turn allows for a greater calcium ion extraction from the solution. As a result, the calcium phosphate precipitates multiply on the surface. However, unlike crystalline forms such as anatase and rutile, the presence of amorphous oxides on the titanium surface prevents the nucleation of calcium phosphate [63]. The diminished peak intensity and the elimination of Ca and P signals in the XRD data can be partially attributed to the lower nucleation capacity of the amorphous oxides.

Ca in the surface coating slowly dissolving and passing through the surrounding solution causes a rise in pH and an increased concentration of Ca ions beyond the titanium surface. The change in the chemical composition promotes the formation of calcium phosphate nuclei. However, the diminished peak intensity of Ca signals in the XRD data could be partially attributed to the dissolution of Ca from the surface coating [64].

It is believed that hydrothermal treatment duration influences hydroxyapatite (HA) formation [65], which is crucial for the aforementioned reasons. Consequently, it is conceivable that the duration of the experimental treatment has an effect on the XRD results.

### 3.3. SEM, OM, and EDS

The mapping of chemical elements on the surface of deposited films by micro-arcing is shown in Figure 6, where the image is taken for 0-, 1- and 2-pass Ti substrates with a magnification of 3000. The deposited films are multiple layers and are obviously porous. The pore size and density are quite the same, which means the ECAP-processed Ti substrate has little influence on the deposited films by micro-arc.

Several other studies have investigated the microstructure and properties of TiO_2_ coatings produced by MAO. The coatings exhibited a mixed-phase structure of anatase and rutile, which is not consistent with the findings of this study [66]. One investigation [67] demonstrated that the TiO_2_ coatings produced by MAO exhibited a better corrosion resistance than the bare Ti substrates, while another study [55] found that the coatings exhibited good mechanical properties and wear resistance. Research results [57] showed that the TiO_2_ coatings produced by MAO could potentially be used as protective coatings for titanium alloys, and a study [41] found that the coatings exhibited good biocompatibility, making them suitable for use as implant coatings. Finally, another study [67] demonstrated that the TiO_2_ coatings produced by MAO exhibited good corrosion resistance and wear resistance, further supporting the potential use of such coatings in various applications.

Taken together, these studies demonstrate the versatility and potential of TiO_2_ coatings produced by MAO in various fields, including corrosion resistance, mechanical strength, wear resistance, and biocompatibility. The findings of this study contribute to the growing body of research on the properties and potential applications of MAO Ti coatings, highlighting the need for further investigation and development of such coatings in various fields.

To further identify the composition of the films, we examined the elemental mapping in the area of these SEM images. Figure 7 shows the mapping results for each Ti substrate, and three important elements, namely, phosphorus, calcium, and carbon, are clearly found. In other words, the deposition by micro-arc is successful in delivering these elements from the following aqueous electrolytes: calcium glycerophosphate hydrate (C_7_H_7_CaO_6_P·2H_2_O) and calcium acetate (Ca(CH_3_COO)·2H_2_O). The EDS mapping is conducted for all three passes with the micro-arc oxidation coating applied, namely, 1-pass, 2-pass, and 3-pass of titanium specimen.

However, since XRD does not present any crystalline phase containing or related to these elements, this means that these films are most likely to be amorphous.

The results of a study that applied both techniques to titanium specimens have been presented in this paper, and the elemental mapping and XRD analysis of the coated specimens were discussed.

The elemental mapping results presented in Figure 7 clearly show the successful deposition of three important elements, namely, phosphorus, calcium, and carbon, onto the titanium substrate using the micro-arc oxidation technique. This observation is consistent with previous studies that have reported the ability of MAO to deliver various elements onto the surface of metals through the electrolyte solution used during the process [68]. The presence of these elements on the surface of the titanium substrate is expected to improve its biological properties, such as osseointegration, which is important for biomedical applications [18].

It is worth noting that the absence of any crystalline phase containing or related to these elements in the XRD analysis suggests that the films are most likely amorphous. This observation is consistent with previous studies that have reported the formation of amorphous films on titanium substrates using the MAO technique [69,70,71]. The amorphous nature of the films can have both advantages and disadvantages. For example, amorphous films can have improved mechanical properties, such as hardness, compared to crystalline films [72,73]. On the other hand, the stability and long-term behavior of amorphous films are not well understood, and they can be prone to degradation over time [74].

In summary, the results of this study indicate the successful deposition of important elements onto the surface of titanium substrates using the micro-arc oxidation technique. The films were found to be amorphous, which can have both advantages and disadvantages depending on the application. Further studies are needed to fully understand the long-term behavior and stability of these films.

Optical digital microscopy (OM) reveals that grain dimensions decreased during plastic deformation, resulting in an increase in grain boundaries for a 2 pass ECAPed titanium sample.

Compared to the as-received sample of titanium, the uniformity of particle refinement was substantially enhanced after two cycles of ECAP. Deformation also appears to have increased the number of defects in the material, as evidenced by the presence of numerous grain boundaries surrounding refined grains. The presence of multiple particle boundaries increases corrosion resistance.

The quantitative results of the element compositions in these mapping areas are presented in Figure 8a for the average weight percentage of each mapped element. The major elements in the area, as expected, are Ti and O. Both account for slightly more than 80%. The rest are carbon, phosphorus, and calcium. Among these three, calcium is more than the other two because the two electrolytes in micro-arc all have calcium as their main constituent.

#### Pore Size

Utilizing ImageJ software(ImageJ 1.51, National Institutes of Health (NIH), Bethesda, MD, USA) to analyze the microstructure SEM image, a graph was generated to depict the frequency of ferret diameter for commercially pure titanium with 0 pass, 1 pass, and 2 pass. The obtained results indicate that the average diameter was 0.899, 0.746, and 0.735 μm, respectively, as shown in Figure 8b.

The main objective of this study was to investigate the influence of pass number on the microstructure of commercially pure titanium. The findings of this study revealed a decline in the average ferret diameter as the pass number increased. This result is in line with the previous research, which reported a decrease in grain size with an increase in the pass number during the cold rolling process of titanium [75].

However, in contrast to the study, which reported an increase in grain size with an increase in the pass number, the present study exhibited contradictory results. This could be attributed to the differences in processing techniques or the purity of the titanium used in the study [76]. Furthermore, the microstructure of the 2-pass sample revealed a more refined structure than the 0-pass sample. This observation is consistent with the research, which indicated that multi-pass rolling leads to a refined microstructure and improved mechanical properties of titanium [77].

### 3.4. Tafel Plot

Figure 9 represents the results of a potentiodynamic Tafel plot analysis conducted on commercially pure titanium samples with varying degrees of coating and passes. The Tafel plot technique is used to study the corrosion behavior of metallic materials by plotting the logarithm of the corrosion current density against the electrode potential. The potential values in the table represent the electrode potential values measured during the analysis.

The plot is made by measuring the anodic and cathodic currents on a logarithmic scale as a function of the applied potential. The slope of the Tafel line gives information on the kinetics of electrochemical reactions, and the intersection of the anodic and cathodic Tafel lines gives information about corrosion characteristics. The formula below can be used to determine the corrosion parameters from the Tafel plot.

From the results presented in Figure 9, it can be observed that the electrode potentials for the uncoated samples are relatively lower than those of the coated samples. The uncoated samples show a gradual increase in electrode potential as the number of passes increases, from 1.12 V for 0pass to 1.17 V for 1-pass and 1.19 V for 2-pass, indicating a gradual decrease in the corrosion rate as the number of passes increases. This trend is expected because the more the metallic surface is covered, the less susceptible it is to corrosion.

On the other hand, the coated samples show a significant increase in electrode potential values compared to the uncoated samples. The 0-pass coated sample has an electrode potential of 1.42 V, which is significantly higher than that of the uncoated 0-pass sample (1.12 V). The electrode potentials for the 1-pass and 2-pass coated samples are 1.44 V and 1.47 V, respectively; thus, indicating that further coating does fairly provide additional protection against corrosion. This suggests that the coating is highly effective in reducing the corrosion rate of the CP Ti sample.

The results of the Tafel plot analysis suggest that the coating significantly reduces the rate of corrosion compared to the uncoated samples. The increase in electrode potential values of the coated samples indicates that the coating provides a more stable and passive layer, preventing the metallic samples from further corrosion. The effect of additional coating passes appears to be limited, with no significant change in electrode potential observed between 1-pass and 2-pass coated samples. Overall, the results suggest that the coating significantly enhances the corrosion resistance of the metallic samples. The findings could be useful in designing and developing more effective corrosion-resistant coatings for industrial applications.

When testing the titanium samples for corrosion behavior in a 3.5% NaCl electrolyte solution, different variables contribute to the reported decrease or increase in corrosion potentials.

Titanium, when exposed to an electrolyte, has a higher initial corrosion potential than other metals. As a result of the formation of a passive oxidation layer on its surface during the corrosion process, titanium’s corrosion potential decreases. By operating as a barrier and reducing the rate of corrosion, this inert oxide layer prevents further deterioration. Micro-arc oxidation (MAO) enhances the corrosion resistance of titanium by producing a ceramic oxide layer on its surface. The MAO coating enables barrier protection, reduced diffusion of corrosive species, and enhanced passive oxide layer stability. As a result, the titanium’s corrosion potential is decreased relative to both its as-received state and that of titanium that has been substantially deformed.

Furthermore, a CP Ti sample that has been severely plastically deformed may develop dislocations and grain boundaries during the deformation process, which may result in localized corrosion. The deformed structure may provide favorable corrosion initiation sites, thereby increasing the titanium’s corrosion susceptibility relative to its as-received state. Using equal-channel angular pressing, it is claimed that the grain size can be refined, resulting in enhanced corrosion resistance [78,79,80]. Figure 9 demonstrates that these variations in corrosion potential are negligible, indicating that two passes of ECAP on a titanium sample are inadequate. Despite this, the study’s findings provided essential information on how grain variation influences corrosion resistance. This effect could account for the observed decrease or increase in corrosion or passivation potential.

#### Corrosion Loss

The data presented in Table 1 provide a view of the effectiveness of coatings in reducing the corrosion rate of commercially pure titanium samples. This finding is consistent with previous research studies that have examined the impact of coatings on metallic materials. This experiment conducted a similar study on a titanium alloy and discovered that the corrosion resistance of the alloy significantly increased with the use of a protective coating [81].

In their study, the coating’s thickness and composition had a significant effect on the metal corrosion resistance. However, the effectiveness of the coating in reducing the corrosion rate of the metal depends on the selection of the coating material and method of application. Discovering the most effective coatings for various metallic materials and applications requires additional research.

The corrosion loss, expressed in miles per year (mmpy), was used in Equation (1) to calculate the corrosion resistance of the samples. The samples were labeled as follows: 0Pass (untreated), 1Pass (after one ECAP pass), 2Pass (after two ECAP passes), 0PassMAO (after MAO treatment), UFG1MAO (after 1-pass ECAP treatment followed by MAO treatment), and 2PassMAO (after 2-pass ECAP treatment followed by MAO treatment).

According to the corrosion loss measurement data in Table 1, 2PassMAO had the lowest corrosion rate (0.495 mmpy), followed by 1PassMAO (0.502 mmpy), 0PassMAO (0.510 mmpy), 2Pass (0.610 mmpy), 1Pass (0.649 mmpy), and 0Pass (0.673 mmpy). These findings indicate that increasing titanium’s corrosion resistance can be accomplished by combining ECAP and MAO surface modifications.

For corrosion analysis, electrochemical impedance spectroscopy (EIS) was utilized. The modified equivalent circuit of Randles was then used to match the gathered impedance data. This circuit model shows different mechanisms at the sample–interface contact. In this investigation, corrosion was analyzed using conventional electrical circuits that are comparable and substantiated by physical evidence. These circuits are illustrated in Figure 10a–c. In accordance with Table 2, the as-received, 0PassMAO, 1Pass, and 2Pass samples exhibit the same corrosion behavior as the sample with a natural oxide coating (passive TiO_2_ layer), as depicted in Figure 10a. Due to the diffusion restriction in the corrosion process, a Warburg resistance (W) was applied to the pure titanium sample coated with amorphous hydroxyapatite (HA) in the circuit depicted in Figure 10b. The circuits in Figure 10c describe the passive TiO_2_ layer and the MAO-coated double layer, respectively, as Rct with the lowest value and R3 with a greater value in cm^−2^ are introduced (refer to Table 2).

The Rs represents the solution or electrolyte resistance, while R_ct_ represents the electron-transfer resistance; a constant phase element (CPE_1_) represents the real double-layer capacitance (C_dl_), and a constant phase element (CPE_2_) represents the Warburg impedance (Z_w_). The electron transfer rate constants (k_et_) were computed as follows [82,83]:(5)$$k_{et}=\frac{1}{2R_{ct}{CPE}_{2}}$$

The results in Table 2 show that the electron transfer rate constant, k_et_, is larger for the 0Pass, 1Pass, and 2Pass samples than for the ECAP- and MAO-treated samples, with the exception of the 2PassMAO sample, which also has a high value of k_et_. This is due to the lower R_ct_ (charge transfer resistance) values seen in these samples. As a result, lower R_ct_ values lead to higher k_et_ values, suggesting a more efficient electron transfer mechanism in the samples mentioned.

From Figure 10c, R3 and CPE_2_ are parallel-connected elements that characterize the charge transfer process at the nonporous layer/electrolyte interface by measuring the charge transfer resistance and capacitance of the electric double layer, respectively. The following equation defines a constant phase element (CPE), which represents the capacitive element [82,84].
(6)$$Z_{CPE}=\frac{1}{Y_{0}({j\omega )}^{n}}$$

It is evident that constant $Y_{0}$, (2πf) radial frequency ω, and exponent n are all related. Exponent n is derived from the logarithmic plot of impedance (Z) versus frequency (f). When n = 0, the constant phase element (CPE) functions as a pure resistor, when n = 1 as a pure capacitor, and when n = −1 as an inductor. When n = 0.5, the Warburg impedance (Zw), which is related to mass transport control due to ion diffusion at the electrode-solution interface, is present. CPE is determined by electrode properties (e.g., roughness), relaxation durations at the electrode–electrolyte interface, porosity, and dynamic disturbance associated with diffusion [85]. In the case of the 0Pass and 2Pass ECAPed samples, when the value of n1 approaches 1, a flawless capacitive behavior is indicated by Table 2. However, because n2 ranges from 0.52 to 0.91 (instead of the Warburg diffusion), the electrode’s adsorbed coating is porous. Using Equation (5), the electron transfer rate constant (ket) can be calculated. Bode diagrams (phase angle versus log f, Figure 10b) reveal that the phases are not 90 degrees, as predicted for optimum capacitive behavior, but rather from 50 to 83 degrees. As evidenced by the gradients of the Bode diagrams (log Z vs. log f, Figure 10a), logarithmic plots of impedance (Z) versus frequency (f) reveal an unchanging slope of −0.78 in the middle-frequency band, exhibiting pseudo-capacitive behavior. At high frequencies, gradients approaching 0 indicate resistant behavior. This is due to the coating on the HA titanium samples’ composition and structure, which makes it excellent for storing and discharging charge during electrochemical operations.

Figure 10c demonstrates the absence of Warburg impedance (W = 0), but Figure 11 presents experimental data fitted using the same circuit. It properly describes the experimental data and has an approximation error of roughly 3%. In contrast to previous research [82], fitting the 1PassMAO using the equivalent circuit shown in Figure 10b did not result in a significant drop in the fitted result when the Warburg element was included, according to the scheme shown in Figure 11b. The experimental data contain a 4% approximation error. This variance might be caused by a number of factors, such as the experimental apparatus, substrate used, electrolyte solution used, and so on.

The charge transfer resistance (R_ct_) at low frequencies is significantly increased by the creation of an MAO coating on the 0Pass, 1Pass, and 2Pass samples, as shown by the EIS data (Figure 11b, Table 2).

Both Figure 11b and Table 2 show that the 0PassMAO and 1PassMAO samples also show a rise in R_ct_ at low frequencies, but to an even greater extent, around 10^3^ times that of Rs, showing the establishment of a second layer due to the HA coating applied by MAO. The 0Pass, 1Pass, and 2Pass samples exhibit charge transfer resistance values between 58 and 111.72 Ω·cm^−2^, whereas the 0PassMAO and 1PassMAO samples vary from 3381.64 to 6187.4 Ω·cm^−2^. It is interesting to see that severe plastic deformation and MAO coating seem to have an effect on the charge transfer resistance. Hence these data demonstrate that ECAP technology improves the titanium sample’s resistance to passivation. The polarization resistance of the double layer may be affected by the introduction of a new resistance component, R3 (Rp) polarization resistance, in the 2PassMAO sample (Figure 10c). In contrast to the exterior porous coating, which has a resistance value of 6187.4 Ω·cm^−2^, the inner oxide layer, which functions as a barrier layer, has resistance values of approximately three orders of magnitude greater. Table 2 shows that the capacitance of the double layer is drastically decreased by hundreds when MAO coating is applied to the 2PassMAO sample, while the resistance is increased by three orders of magnitude. It is also important to note that the charge transfer resistance (Rct) for the 1Pass and 2Pass samples is increased by a factor of 1.5 compared to the as-received 0Pass sample.

Numerous studies have examined the effect of ECAP and MAO surface modification on the corrosion resistance of Ti and its alloys. For example, after the ECAP treatment, the grain size of pure Ti was refined and the density of dislocations was increased, resulting in a substantially improved corrosion resistance [86]. Additionally, the ECAP treatment and subsequent MAO surface modification improved the corrosion resistance of the Ti-6Al-4V alloy [75,86].

Figure 11b demonstrates that the phase angle of the sample treated with 2-pass ECAP and MAO is greater at lower frequencies. A larger phase angle indicates that the system exhibits a greater capacitance. This type of behavior has been linked to the formation of a barrier layer on the surface of the substrate, in this instance, the 2-pass ECAP and MAO-treated sample. By preventing the passage of charge, this protective coating significantly slows down the corrosion process. Thus, a larger phase angle at low frequencies is frequently indicative of a more effective and protective coating. Unlike the UFGMAO sample, due to its higher frequency and lower phase angle, the CGMAO sample is more susceptible to corrosion. Figure 11a depicts the Bode graphs; the absolute value of Z in UFGMAO titanium is greater at lower frequencies. Interpreting this behavior, a greater absolute value of Z at lower logarithmic frequencies indicates an increased impedance or corrosion resistance. This suggests that the formation of a protective oxidation layer on the surface of the material is responsible for the increased corrosion resistance observed at lower frequencies in titanium treated with two passes of ECAP and MAO. This oxide layer acts as a barrier, reducing corrosion by isolating the metal from its corrosive environment.

The Nyquist plot depicted in Figure 11c,d (zoomed area selected in blue dashed square box) enables us to conclude that the UFGMAO sample has the highest value of Z imaginary in ohms per centimeter square. Compared to the other samples, this indicates a reduced corrosion rate. It also indicates that MAO and multi-pass ECAP have the potential to enhance CP Tis corrosion resistance. Figure 11c indicates that the as-received sample is more corroded than the other samples, indicating a higher corrosion rate.

It has been hypothesized that MAO surface modification increases the corrosion resistance of Ti and its alloys via the formation of a protective oxide layer on the surface of the material. The MAO treatment generates surface fissures containing Ca-P minerals that promote the formation of bioactive hydroxyapatite [87]. The hydroxyapatite layer provides a stable and protective contact between the material and the corrosive environment by acting as a barrier to prevent the diffusion of corrosive ions.

The chi-square value (χ^2^) lies within the range from approximately 5.08 × 10^−5^ to 2.49 × 10^−3^, indicating that the experimental results are consistent with the results of the fitting (see Table 2). This result demonstrates that the appropriate electrical circuit models were selected for the investigation.

By referring to all the results from the corrosion test, we can roughly report that the impact of equal-channel angular pressing (ECAP) on the corrosion resistance of the titanium samples used in this investigation cannot be overemphasized. Figure 6a demonstrates that, in comparison to the as-received sample (0Pass CP Ti), the titanium sample exposed to a 2-pass ECAP underwent substantial deformation, resulting in a significant reduction in the grain size and the formation of multiple grain boundaries and defects.

When tested for corrosion behavior, the corrosion resistance (case of a double layer with MAO HA coating applied) and passivation resistance (with a natural coating of TiO_2_) of 0-pass, 1-pass, and 2-pass samples without micro-arc oxidation (MAO) coating all increase. The corrosion resistance of ECAPs ultrafine-grained (UFG) titanium exceeds that of coarse-grained (CG) titanium, as demonstrated by testing with simulated saline [88]. According to a number of studies, the corrosion resistance of UFG/nano-crystalline (NC) titanium is superior to that of titanium in its coarse-grained state. The reduced particle size facilitates the formation of passive coatings on the surface, which in turn improves their adhesion to the crystallographic defects, resulting in enhanced corrosion resistance. Due to the small particle size and numerous grain boundaries, magnesium (Mg) metal is especially susceptible to corrosion.

In conclusion, the ECAP procedure refines titanium’s grain and forms grain boundaries and defects, thereby increasing the metal’s resistance to corrosion. The phenomenon of enhanced corrosion resistance, which is supported by earlier investigations of UFG/NC titanium compared to CG titanium, has been attributed to the faster production and better adhesion of passive coatings on surface crystallographic defects.

### 3.5. Biomedical Experiment

At the National Taiwan University Hospital, researchers studied cell proliferation in four distinct sample materials in an in vitro investigation. In our cytotoxicity test setting, the following methods were chosen according to the ISO 10993-5(2009) indication (ISO 10993-5, 2009) [89]. Significantly lower absorbance values were seen on day 3 for cells grown on CG and CG with MAO surfaces compared to cells grown on UFG and UFG with MAO surfaces on day 1, as shown in Figure 12a. Furthermore, on day 7, there were substantial variations in absorbance across CG and UFG surfaces. This study drew conclusions regarding the difference in cell viability between groups treated for 1, 3, and 7 days using one-way ANOVA followed by post hoc multiple comparison analysis. The post hoc multiple comparison will only be conducted if the Significance value (*p* value) is less than the alpha value, 0.05. In addition, the HSD Tukey test assumed normal distribution (refer to Table 3, Table 4, Table 5, Table 6, Table 7 and Table 8). In contrast to previous studies [90], we utilized both human pulp cells and human dental periodontal cells to evaluate cell viability. Consequently, our findings offer novel and significant scientific insights.

As depicted in Figure 12b, another significant result involved human periodontal ligament cells, where the number of cells on UFGMAO was significantly higher than on UFG after 1 and 7 days, with similar numbers observed after 3 days. Human pulp cells underwent a three-day viability test with four titanium samples (CG, CGMAO, UFG, and UFGMAO) and a TCP control group. On each of the three sampling days, the CG and UFG samples had lower pulp cell viability than the TCP control group, as shown in Figure 12a.

On day 7, there was no difference in pulp cell viability between UFG and UFGMAO samples and CG and TCP. Using micro arc oxidation (MAO) technology and mineral solutions containing Ca and P to modify commercially pure titanium samples (CGMAO and UFGMAO) may improve pulp cell viability; however, additional research is necessary to optimize processing procedures and improve biocompatibility for medical implants. This study may appear to contradict other research, but it contributes to the corpus of scientific knowledge on the subject.

Human dental pulp cells (hDPCs) were statistically examined at 1, 3, and 7 days after CG treatments using SPSS version 27 software. We used a one-way analysis of variance (ANOVA) to compare group averages across time, followed by the Tukey Honestly Significant Difference (HSD) test if the ANOVA result was statistically significant (*p =* 0.05). In scientific research, the analysis of variance (ANOVA) and Tukey’s honestly significant difference (HSD) tests are widely used to compare group averages and find statistically significant group comparisons. Differences in viability were found in our research by comparing the viability of CG on Day 1 to CG on Days 3 and 7, CG on Day 3 to Days 1 and 7, and CG on Day 7 to Day 1 and 3. All of these *p*-values were less than 0.05, showing statistically significant variations in cell viability between time periods.

This finding provides insight on the time-dependent dynamics of CG viability following hDPC treatment and shows that CG viability changes dramatically over time. Possible reasons for observed viability variations include HDPC metabolic activity, cellular proliferation, and interactions between hDPCs and the CG environment. Our results in Table offer clarity on the temporal dynamics of CG viability as well as the possible therapeutic effects of hDPCs in CG regeneration.

ANOVA and a post hoc Tukey HSD test were utilized to gain insight into how the duration of exposure of the CG sample to human periodontal cells impacted its viability. We were able to test the null hypothesis that all population means are equal by comparing the means of multiple categories using ANOVA. Table 5 contains the outcomes of a one-way analysis of variance (ANOVA) that was followed by a post hoc test conducted under the assumption of equal variance to further analyze and comprehend the ANOVA outcomes. The post hoc Tukey HSD analysis (refer Table 6) enabled pairwise comparisons and enabled a more comprehensive evaluation of the variations in CG sample viability on days 1, 3, and 7. According to the data, there was no statistically significant difference in survival rates between CG at days 1 to CG at Day 3 (*p* > 0.05). Comparing the CG sample from day 1 to day 7 (*p* < 0.05), a statistically significant and substantial change was observed. The differences in viability between day 3 compared to day 7 were also statistically significant. A consistent pattern emerged when comparing the CG sample from day 7 to those from days 1 and 3. The time-dependent viability of the CG sample when exposed to human periodontal cells was investigated further using ANOVA and the post hoc Tukey HSD test, providing additional evidence for the complex dynamic of cell viability in these experimental conditions.

On day 1, human periodontal cells and human dental pulp cells were used to assess the differences in cell viability between the BTI (control group), CG, CGMAO, UFG, and UFGMAO treatment groups. Using the assumption of equal variances, we conducted a one-way analysis of variance (ANOVA) on variations in cell viability, followed by post hoc multiple comparisons using the Tukey Honestly Significant Difference (HSD) test. According to Table 7, there were statistically significant differences in cell viability on day 1 after treatment with periodontal cells between the aforementioned groups. To assess the differences in first-day cell viability between groups, we performed a post hoc multiple comparison using the Tukey HSD test. There were no statistically significant differences in cell viability between groups treated on day 1 with human dental pulp cells. These groups’ ANOVA results were insignificant (*p* > 0.05). Therefore, post hoc multiple comparisons were not performed on the groups that were treated with dental pulp cells. When comparing the viability of the CG sample to that of the CGMAO, UFG, UFGMAO, and BTI groups, there were no significant differences (Table 8). These results pertain to periodontal cells that were treated for the same duration. Thus, the null hypothesis that there was no difference between the CG sample and the other groups on day 1 could not be rejected. After 1 day of exposure to periodontal cells, there was no statistically significant difference in cell viability between the CG sample and the other treatment groups. This demonstrates that the treatment condition has no significant effect on the survival of CG samples over this time period. The treatment condition in our study is an example of mixed factor data, and the use of one-way ANOVA followed by post hoc multiple comparisons, as implemented in this study, is a robust and legitimate statistical strategy for evaluating such data. Using this method, we determined that there were no statistically significant differences between the CG sample and the other groups in terms of cell viability on day 1, and that there were substantial differences between the periodontal cell treatment groups. This technique enhances the reliability and precision of our data by enhancing our understanding of how different cell types and therapies influence cell viability.

Our findings indicate that cell viability assays are valuable for determining the effect of periodontal cells and dental pulp cells on the overall health of CG, CGMAO, UFG, and UFFG samples. However, additional research is required to determine the therapeutic significance of these results and to comprehend the mechanisms underlying the observed differences in viability. When evaluating cell viability data under mixed factor designs, one-way ANOVA followed by post hoc multiple comparisons is an effective statistical technique. Our findings indicate that there are substantial differences in cell viability between groups treated with periodontal cells on days 1, 3, and 7. These results provides an understanding of how different cell types and treatments affect cell survival.

The success or failure of medical implants depends on their biocompatibility, or their ability to integrate with the body. Micro arc oxidation (MAO) technology was used to modify commercially pure titanium (CG and UFG) with mineral solutions containing Ca and P (CGMAO and UFGMAO), and then the materials’ ability to support the viability of periodontal cells was evaluated. The periodontal cell viability in all titanium samples and the BTI control sample decreased over time, with the CG and UFG samples consistently demonstrating lower vitality than the BTI control sample over the course of seven days. On days 3 and 7, the periodontal cell viability of the CGMAO and UFGMAO samples were lower than that of the BTI control group as depicted in Figure 12b. In terms of periodontal cell survival, the CGMAO and UFGMAO samples outperformed the unmodified CG and UFG samples on day 1, but by days 3 and 7, the difference had dissipated. On day 7, the periodontal cell viability in the CGMAO and UFGMAO samples were greater than that of the CG and UFG samples, but lower than that of the BTI control group.

In this study, non-crystalline HA was synthesized using PEO technology, and it was demonstrated that amorphous coatings of titanium and hydroxyapatite oxides possessed low biological activity. However, research indicates that it still has certain beneficial consequences. Amorphous calcium phosphate (ACP) is more easily absorbed and digested by living organisms than crystalline calcium phosphates such as hydroxyapatite (HAP), leading in the creation of new bone tissue. On ACP substrates, more osteogenic cells adhere and proliferate, indicating higher bioactivity [91]. According to in vitro studies, amorphous calcium phosphate coatings promote quicker bone growth and greater cell differentiation than crystalline phosphate coatings [92]. Plasma solidification results in an unstructured substance known as calcium amorphous state, which is a frozen liquid rather than crystalline phosphate. This substance dissolves quicker than crystalline phosphate and initiates the bone apposition process in osteoblasts. The non-crystalline calcium phosphate layer also initiates the bone apposition process of osteoblasts [93].

Our findings indicate that the periodontal cell viability can be enhanced on samples of commercially pure titanium by modifying the surface with MAO technology and mineral solutions, including Ca and P. Further research is required to optimize titanium refining procedures for greater biocompatibility in medical implants, as periodontal cell survival on all titanium samples remained inferior to that of the BJTI control group. Our research provides information regarding the biocompatibility of titanium samples intended for use in medical implants. Using cutting-edge MAO and SPD technologies to maximize periodontal cell survival on titanium substrates, titanium implants have the potential to improve patient outcomes if these techniques can be optimized through further research.

## 4. Conclusions

Micro-arc oxidation (MAO) technology, mineral solutions containing Ca and P, and the equal-channel angular pressing (ECAP) method for severe plastic deformation (SPD) can be used to improve the mechanical properties, corrosion resistance, and cell viability of CP Ti, which were observed through this study. The following findings of this study could be drawn:Despite the fact that the anticipated equivalent strain for 2 passes is less than 1.5, which is inadequate to produce equiaxed evenly deformed ultrafine grain CP Ti, the influence of equal-channel angular pressing (ECAP) on the corrosion resistance of titanium samples still, to some extent, remains noticeable;The presence of amorphous oxides prevents calcium phosphate nucleation, resulting in diminished peak intensity and elimination of Ca and P signals in XRD data;The surface modification process using MAO and mineral solutions containing Ca and P can change the composition and shape of the surface of pure titanium;The corrosion resistance of commercially pure titanium can be enhanced by integrating ECAP and MAO surface modification;Human pulp and periodontal cell viability can be enhanced by modifying commercially pure titanium samples (0PassMAO, 1PassMAO, and 2PassMAO) with micro-arc oxidation (MAO) and ECAP technology; however, additional testing methods must be developed for improved biocompatibility in medical implants.

Overall, the results indicate that ECAP and MAO may be combined to enhance the mechanical properties of purified titanium, which is beneficial for biomedical applications. Future biomedical engineering research may benefit from the findings because it provides understanding on the behavior of dental pulp cells and periodontal cells. Therefore, the findings of this study will contribute to the development of cutting-edge biomedical implants with enhanced functionality for extended durations.

## Figures and Tables

**Figure 1 micromachines-14-01541-f001:**
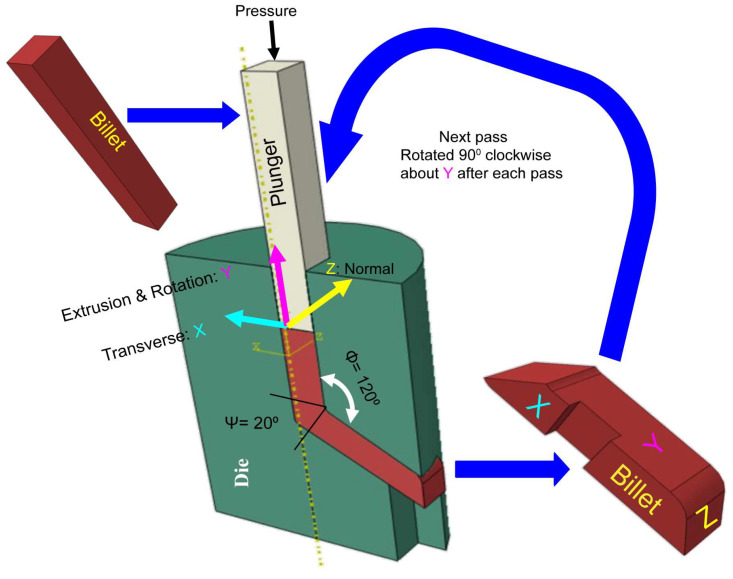
Design and process of the ECAP for grade 4 pure titanium strips.

**Figure 2 micromachines-14-01541-f002:**
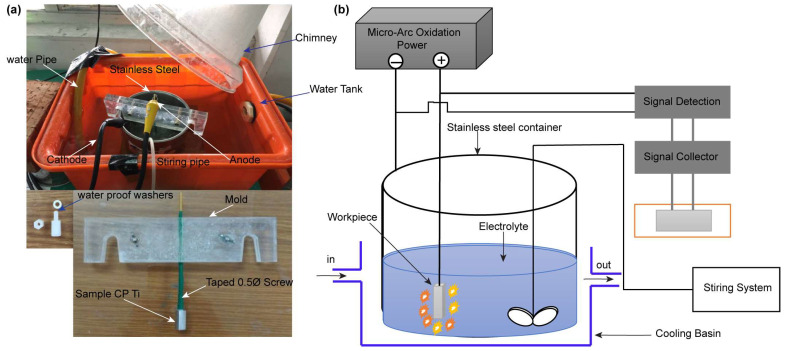
Schematic representation of (**a**) the micro-arc oxidation (MAO) setup and (**b**) coating technology.

**Figure 3 micromachines-14-01541-f003:**
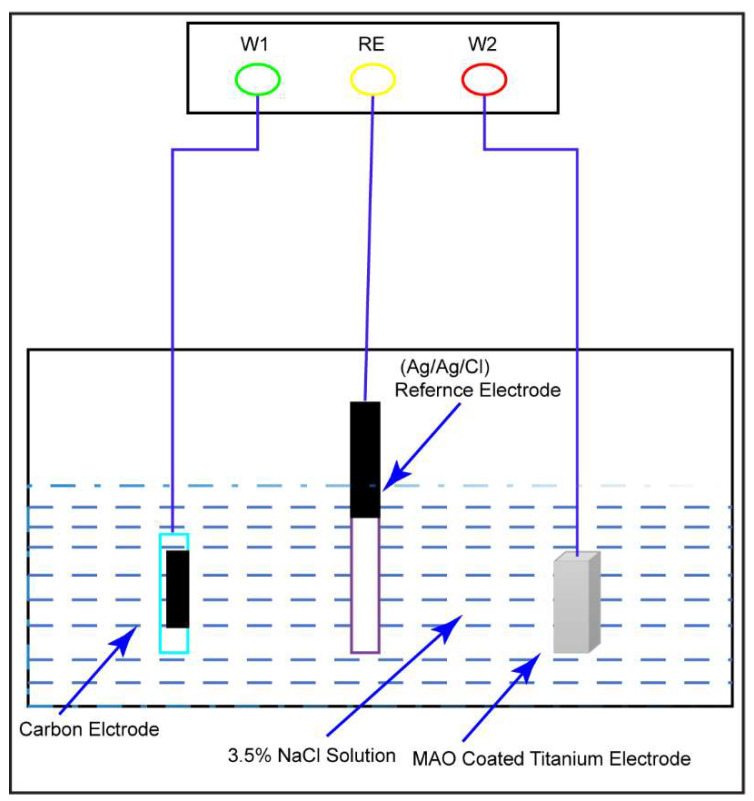
The schematics for potentiodynamic test of untreated, MAO- and ECAP-treated CP Ti.

**Figure 4 micromachines-14-01541-f004:**
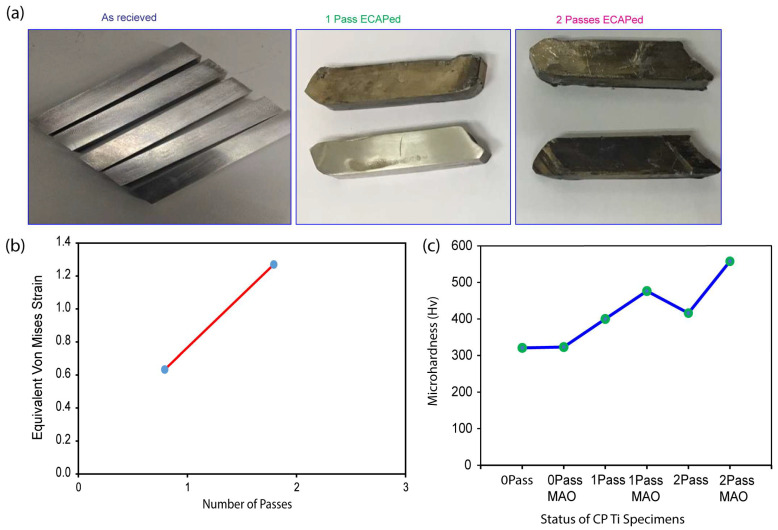
(**a**) CP Ti as-received, 1-pass, 2-pass ECAP treated at 4000C using MoS_2_ lubrication and (**b**) equivalent Von Mises strain against number of passes, and (**c**) microhardness test result of CP Ti samples treated/untreated with ECAP and MAO techonology.

**Figure 5 micromachines-14-01541-f005:**
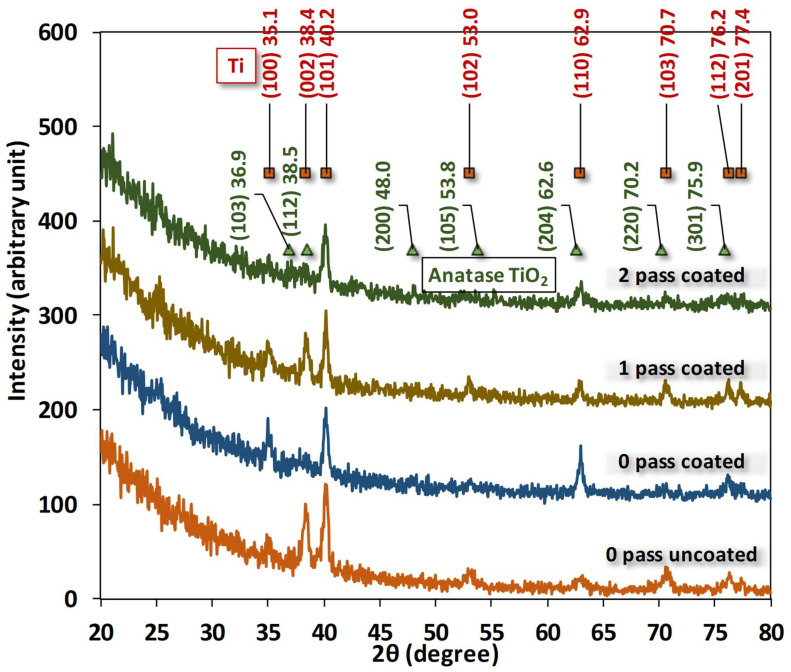
The XRD Image of grade 4 Ti before and after ECAP treatments and micro-arc oxidation for coating.

**Figure 6 micromachines-14-01541-f006:**
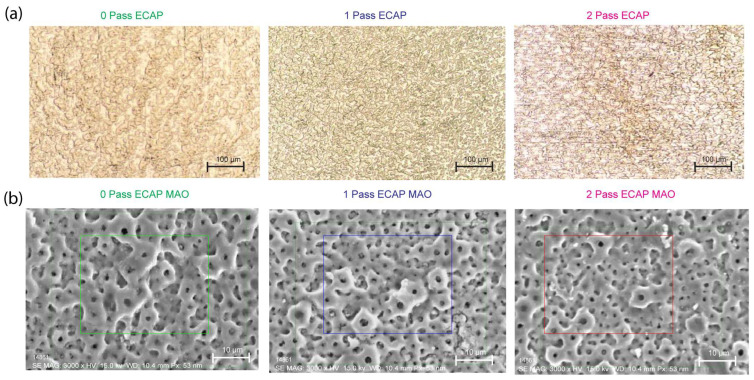
Shows (**a**) the OM and SEM result of grade 4 CP Ti before/after ECAP, and (**b**) the SEM result of CP Ti treated and untreated with ECAP and MAO technology. The green-, blue-, and magenta-colored boxes indicate the location where the EDS mapping is taken.

**Figure 7 micromachines-14-01541-f007:**
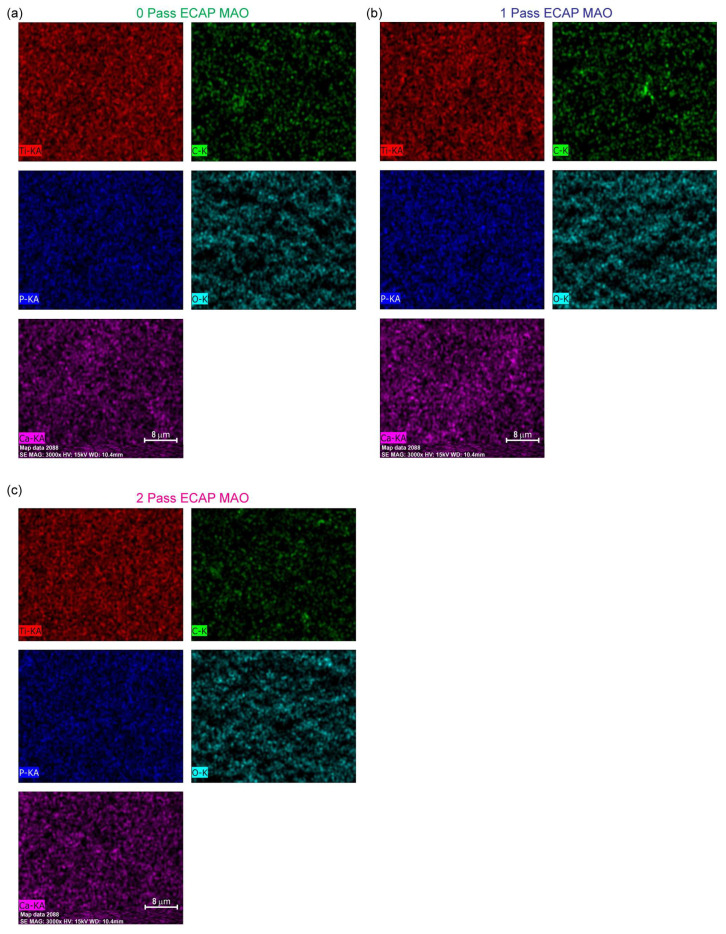
The elemental mapping from the area shown in the SEM image of (**a**) as-received, (**b**) 1-pass ECAP and (**c**) 2-pass ECAP grade 4 Ti after micro-arc oxidation for coating.

**Figure 8 micromachines-14-01541-f008:**
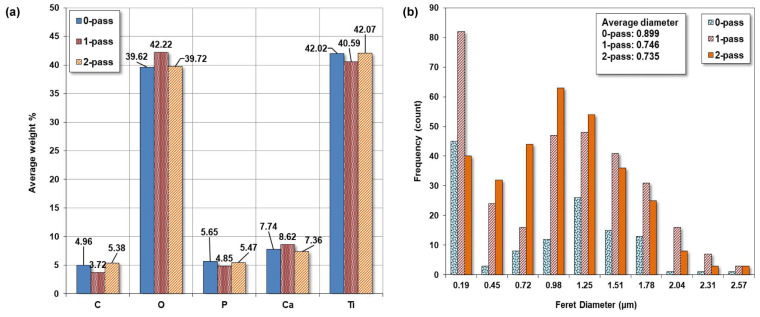
(**a**) The average weight percentage of elements from EDS mapping and (**b**) the pore size of the coating on the samples.

**Figure 9 micromachines-14-01541-f009:**
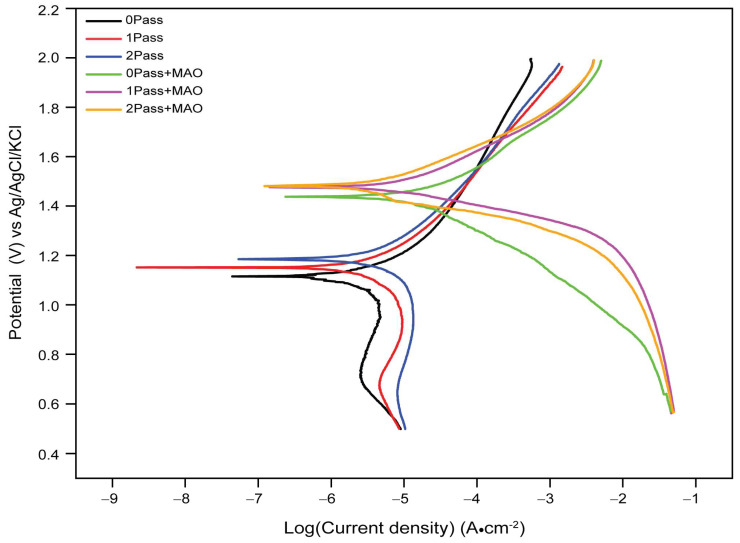
The Tafel plot for open circuit voltage for SEM of grade 4 Ti before/after ECAP and micro-arc oxidation for coating.

**Figure 10 micromachines-14-01541-f010:**
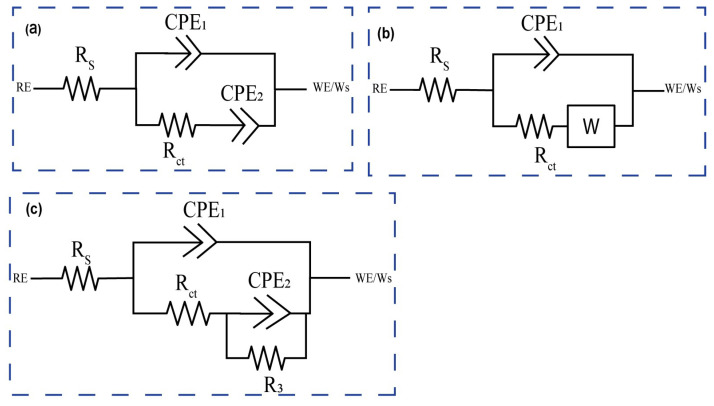
The modified Randles equivalent circuit [82,83] used for fitting the experimental data from EIS test: (**a**) as-received, 0PassMAO, 1Pass, and 2Pass, (**b**) 1PassMAO, and (**c**) 2PassMAO.

**Figure 11 micromachines-14-01541-f011:**
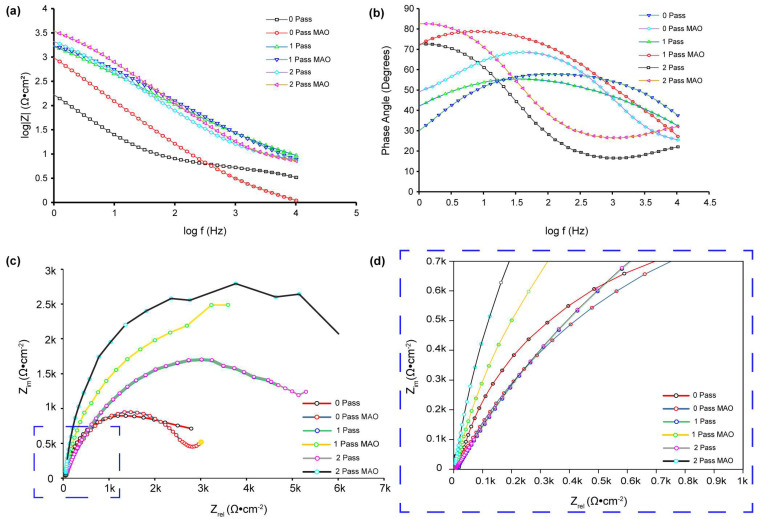
Bode plots, log Z vs. log f (**a**), and phase angle vs. log f (**b**), and Nyquist plot (**c**) and detailed view at lowest Zre and Zim shown in blue box (**d**) of ECAP- and MAO-treated samples in a 3.6% NaCl electrolyte solution. The symbols in (**a**–**d**) represent the experimental data, while the solid lines are the fitted curves using the electrical equivalent circuit models in Figure 10a–c.

**Figure 12 micromachines-14-01541-f012:**
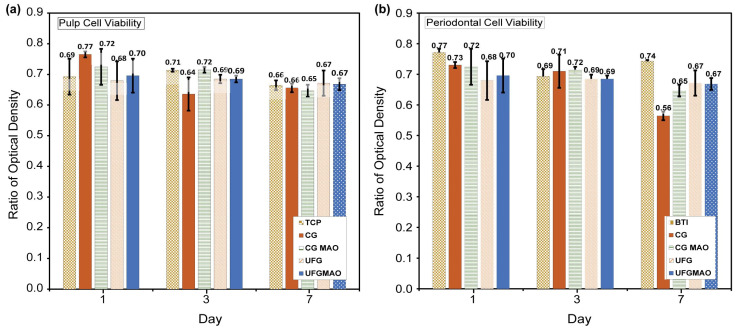
The viability test for (**a**) human pulp cells using the control group TCP and (**b**) human periodontal cells utilized the control group BTI on different titanium samples (CG, CGMAO, UFG, UFGMAO). TCP: Tissue culture polystyrene; CG: 1Pass ECAP without MAO coating; CGMAO: 1Pass ECAP with MAO coating; UFG: 2Pass ECAP; UFGMAO: 2Pass ECAP with MAO coating.

**Table 1 micromachines-14-01541-t001:** Corrosion behavior parameters calculated from the Tafel plot and Stern–Geary equation.

	E_corr_	B_a_	B_c_	R_p_	i_corr_	CR
Samples	(V)	(V)	(V)	$\left(k\Omega \;{cm}^{2}\right)$ (Stern–Geary)	(mA/cm^2^)	mmpy
0Pass	1.12	0.132	0.375	74.734	−0.568	0.673
1Pass	1.17	0.183	0.239	82.230	−0.548	0.649
2Pass	1.19	0.2	0.34	106.311	−0.515	0.610
0PassMAO	1.42	0.256	0.197	112.567	−0.430	0.510
1PassAMO	1.44	0.248	0.192	110.970	−0.424	0.502
2PassMAO	1.47	0.283	0.155	104.169	−0.412	0.495

**Table 2 micromachines-14-01541-t002:** Estimated EIS parameters for surface-modified commercially pure titanium samples in a 3.6% NaCl electrolyte solution.

	Titanium Sample Grain Refinement and Surface Modification Status					
EIS Parameters	0Pass	0PassMAO	1Pass	1PassMAO	2Pass	2PassMAO
R_S/_Ω·cm^−2^	3.08 (4.56)	3.92 (4.48)	7.77 (10.72)	5.43 (3.66)	4.86 (9.72)	1.143 (8.29)
R_ct/_Ω·cm^−2^	58 (4.67)	832.94 (3.86)	87 (2.34)	5343 (1.65)	111 (1.06)	6.96 (4.98)
R_3_/Ω·cm^−2^	-	-	-	-	-	6187.4 (8.79)
CPE_1_/F·cm^−2^	6.64 × 10^−5^ (7.06)	9.12 × 10^−5^ (0.85)	1.14 × 10^−4^ (4.52)	1.53 × 10^−4^ (0.79)	5 × 10^−4^ (7.20)	8.64 × 10^−5^ (3.75)
n_1_	0.65 (1.19)	0.69 (0.17)	0.70 (0.99)	0.64 (0.19)	0.52 (0.76)	0.91 (0.71)
CPE_2_/F·cm^−2^	1.54 × 10^−5^ (5.82)	5 × 10^−4^ (1.61)	2.22 × 10^−4^ (8.27)	-	5 × 10^−4^ (6.45)	1.01 × 10^−4^ (5.48)
n_2_	1 (11.89)	0.043 (16.04)	0.10 (18.93)	-	1 (8.17)	0.89 (1.27)
W_1/_Ω·S^−1^·cm^−2^				142.73 (4.14)	-	-
K_et_/S^−1^	559.79	1.20	25.89		9.01	711.28
χ^2^	2.49 × 10^−3^	2.64 × 10^−4^	5.08 × 10^−5^	1.48 × 10^−3^	2.55 × 10^−4^	5.88 × 10^−5^

The numbers in the brackets are the error percentages predicted from the fitting using the circuit shown in Figure 11a–c.

**Table 3 micromachines-14-01541-t003:** Single Anova on viability of CG sample treated by human Dental Pulp Cells for 1, 3, and 7 Days.

ANOVA					
Viability					
	Sum of Squares	df	Mean Square	F	Sig.
Between Groups	0.026	2	0.013	69.847	0.000023
Within Groups	0.002	9	0.000		
Total	0.027	11			

**Table 4 micromachines-14-01541-t004:** Post hoc Tukey HSD on viability of CG sample treated by human Dental Pulp Cells for 1, 3, and 7 Days.

Multiple Comparisons					
Dependent Variable: Viability					
Tukey HSD					
(I) SampleCG137	(J) SampleCG137	Mean Difference (I − J)	Std. Error	Sig.	95% Confidence Interval
					Lower Bound
CG1	CG3	0.080250 *	0.009576	0.00056	0.05351
	CG7	0.109250 *	0.009576	0.00012	0.08251
CG3	CG1	−0.080250 *	0.009576	0.00025	−0.10699
	CG7	0.029000 *	0.009576	0.035	0.00226
CG7	CG1	−0.109250 *	0.009576	0.000	−0.13599
	CG3	−0.029000 *	0.009576	0.035	−0.05574

(*) The mean difference is significant at the 0.05 level.

**Table 5 micromachines-14-01541-t005:** Single Anova on viability of CG sample treated by human periodontal cells compared to itself for 1, 3, and 7 Days.

ANOVA					
Viability					
	Sum of Squares	df	Mean Square	F	Sig.
Between Groups	0.049	2	0.025	37.491	0.00022
Within Groups	0.004	6	0.001		
Total	0.053	8			

**Table 6 micromachines-14-01541-t006:** Post hoc Tukey HSD on viability of CG sample treated by human periodontal cells compared to itself for 1, 3, and 7 Days.

Multiple Comparisons				
Dependent Variable: Viability				
Tukey HSD				
(I) SamplePCDLCG137	(J) SamplePCDLCG137	Mean Difference (I − J)	Std. Error	Sig.
CG1	CG3	0.020333	0.020957	0.620
	CG7	0.166333 *	0.020957	0.001
CG3	CG1	−0.020333	0.020957	0.620
	CG7	0.146000 *	0.020957	0.001
CG7	CG1	−0.166333 *	0.020957	0.001
	CG3	−0.146000 *	0.020957	0.001

(*) The mean difference is significant at the 0.05 level.

**Table 7 micromachines-14-01541-t007:** Single Anova on viability of CG sample treated by human periodontal cells compared to the group for same duration, day 1.

ANOVA					
Viability					
	Sum of Squares	df	Mean Square	F	Sig.
Between Groups	0.007	4	0.002	6.778	0.007
Within Groups	0.003	10	0.000		
Total	0.010	14			

**Table 8 micromachines-14-01541-t008:** Post hoc Tukey HSD on viability of CG sample treated by human periodontal cells compared to the group for same duration, day 1.

Multiple Comparisons					
Dependent Variable: Viability					
Tukey HSD					
(I) SampleTiDAY1	(J) SampleTiDAY1	Mean Difference (I − J)	Std. Error	Sig.	95% Confidence Interval
					Lower Bound
BTI1	CG1	0.040667	0.013433	0.075	−0.00354
	CGMAO1	0.001000	0.013433	10.000	−0.04321
	UFG1	0.047000 *	0.013433	0.036	0.00279
	UFGMAO1	−0.004000	0.013433	0.998	−0.04821
CG1	BTI1	−0.040667	0.013433	0.075	−0.08488
	CGMAO1	−0.039667	0.013433	0.084	−0.08388
	UFG1	0.006333	0.013433	0.988	−0.03788
	UFGMAO1	−0.044667 *	0.013433	0.047	−0.08888
CGMAO1	BTI1	−0.001000	0.013433	10.000	−0.04521
	CG1	0.039667	0.013433	0.084	−0.00454
	UFG1	0.046000 *	0.013433	0.041	0.00179
	UFGMAO1	−0.005000	0.013433	0.995	−0.04921
UFG1	BTI1	−0.047000 *	0.013433	0.036	−0.09121
	CG1	−0.006333	0.013433	0.988	−0.05054
	CGMAO1	−0.046000 *	0.013433	0.041	−0.09021
	UFGMAO1	−0.051000 *	0.013433	0.023	−0.09521
UFGMAO1	BTI1	0.004000	0.013433	0.998	−0.04021
	CG1	0.044667 *	0.013433	0.047	0.00046
	CGMAO1	0.005000	0.013433	0.995	−0.03921
	UFG1	0.051000 *	0.013433	0.023	0.00679

(*) The mean difference is significant at the 0.05 level.

## Data Availability

Not applicable.

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
