# Peer review of "Improving Pure Titanium’s Biological and Mechanical Characteristics through ECAP and Micro-Arc Oxidation Processes"

_micromachines, 2023, doi:10.3390/mi14081541_

Round 1
Reviewer 1 Report
The manuscript presents some interesting results on combining ECAP and MAO to improve the characteristics of pure titanium for biomedical applications. While the results are encouraging, I believe that the authors need to include several key measurements before the paper can be published.
Major issues
1. The authors state in their title and abstract that ECAP can be used to improve the mechanical properties of CP-Ti. However, they do not show any measurement regarding mechanical properties. I believe that they should include at least hardness and tensile measurements, as well as microstructure images, particularly since they clearly state in Line 383 “The main objective of this study was to investigate the influence of pass number on 383 the microstructure of commercially pure titanium “
2. Regarding microstructural evolution, particularly grain refining, the authors say that 2 120º ECAP passes are sufficient for enhancing the mechanical properties, since they lead to an homogenous shear deformation. They do not show the microstructure or crystallographic texture evolution to support this claim. Furthermore, since they have chosen a 120º processing route, I doubt that the shear strain is enough to lead to any significant grain.
3. In terms of calculating the equivalent Von Mises strain, I suggest the authors to check several papers of RZ Valiev. Usually, a high deformation ECAP route with a 90º angle leads to an equivalent Von Mises strain of 1. A “softer” angle of 120º cannot lead to such a high equivalent strain of over 2.
4. There is a long-standing convention to label ECAP processing routes, based on how the samples are rotated between passes, namely routes A, B and C. Again this can be consulted in Valiev’s papers and the authors should use this convention.
5. EDS is not very accurate to determine the volume fraction of light elements. In particular, it is not recommended to measure carbon. The authors should not discuss about carbon content changes using this technique (see, for example, ASTM E1580 standard)
6. I do not believe that ECAP is leading to a significant improvement in corrosion resistance, as seen in Figure 9. I think that the most important factor to improve corrosion resistance is the MAO surface treatment. In short, the authors should make it clearer how ECAP is improving the final product in any regard.
7. Finally, there are some clean up issues such as using sever instead of severe, using total plastic deformation instead of equivalent strain.
English is generally fine, some minor issues such as those highlighted in issue 7.
Author Response
Dear Reviewer;
Your suggestions allowed us to improve our work and offer additional context for our results. We made significant changes to address issues and enhance the research quality. The final piece of writing has been enhanced as a result of your comments and recommendations, and it is now in a better position to make an impression. We trust that the enhanced work will meet your needs and provide answers to your queries. We much appreciate helping us with the present study. The answer to your comments and corresponding updated information are colored in red for easy identifications.

Reviewer 2 Report
1. To increase the interest of readers, in my opinion, it is also necessary to add the PEO method in molten salts in the introduction. https://doi.org/10.3390/ma15207374
2. P.2.3.2 Formula 3. The Stern Geary formula is used to calculate Rp, which is used for potentiodynamic curve analysis (i.e., Tafel) rather than impedance spectroscopy (EIS).
3. P.3.2 The results obtained should be compared with each other. The intensity of the peaks is relatively low, which indicates that the coating is amorphous. A rather strange result, usually PEO coating has high crystallinity. Need to explain. Why are hydroxyapatite phases not visible? A more detailed analysis of the data is needed.
4. Amorphous coating of titanium and hydroxyapatite oxides have deficient biological activity. What is the benefit?
5. Figure 7. What can be seen from EDS mapping? Figure 8 calcium and phosphorus content is high enough; why are its phases not visible on XRD?
6. Line 408 What potentials are described in this sentence? Corrosion, passivation, repassivation?
7. Figure 9 Why is there a decrease/increase in corrosion potentials? The current scale in the figure should be logarithmic.
8. Table 1 ba, bc, and Rp - calculated incorrectly. Judging by your data, the coatings are superconductors.
9. P. 3.4 It is necessary to compare the results in more detail and describe the effect of surface modification on corrosion properties.
10. Figure 10 10 a - on a scale of y -should be log IZI. The x-scale must be logarithmic. Fig. 10 b - scale y - should be the phase angle. Figure 10 c - Ohm for what area?
11. Figure 10 should show the experimental results as dots and the fitting as lines. Adding the quality of approximation of experimental data and fitting data (x2) is necessary. It is required to present a table with the R, C, and n results and add the equivalent circuit. For example, I suggest using the following article https://doi.org/10.1016/j.corsci.2022.110604
In addition, it is necessary to compare the results of EIS and Tafel.
Author Response
Dear Reviewer;
Your suggestions allowed us to improve our work and offer additional context for our results. We made significant changes to address issues and enhance the research quality. The final piece of writing has been enhanced as a result of your comments and recommendations, and it is now in a better position to make an impression. We trust that the enhanced work will meet your needs and provide answers to your queries. We much appreciate helping us with the present study. The answer to your comments and corresponding updated information are colored in bule for easy identifications.

Round 2
Reviewer 1 Report
I believe the authors have made a great effort to address every comment made by this reviewer and I greatly appreciate it. Thanks for all your hard work, I consider that it is ready for publication in its present form.
Author Response
Dear Reviewer,
we are deeply grateful for your thoughtful review and kind endorsement of our work. Your appreciation motivates us greatly.
Best Wishes.

Reviewer 2 Report
Dear authors.
Corrections have been made. I ask you to check the cited literature and make the required corrections in the reference list.
Author Response
Dear Reviewer,
Thank you for your attention to detail and the helpful advice. We'd like to let you know that we've already communicated with our Assigned Editor, Ms. Nadja Kilibarda, about updating the newly mentioned citations in our manuscript. We have been informed that she would assist us with this procedure throughout the revision stage. We really appreciate your time to review our manuscript.
Best wishes.
